# Cost–Utility Analysis of Supervised Inspiratory Muscle Training Added to Post-COVID Rehabilitation Program in the Public Health System of Brazil

**DOI:** 10.3390/ijerph21111434

**Published:** 2024-10-29

**Authors:** Guilherme Pacheco Modesto, Aline Loschi Soria, Luis V. F. Oliveira, Everton Nunes da Silva, Graziella F. B. Cipriano, Gerson Cipriano, Vinicius Maldaner

**Affiliations:** 1Human Movement and Rehabilitation Graduate Program, UniEvangelica, Anapolis 75083-515, Brazil; guilhermemodesto.ndae@escs.edu.br (G.P.M.); oliveira.lvf@gmail.com (L.V.F.O.); cipriano@unb.br (G.C.J.); 2Health Sciences Program, Escola Superior de Ciências da Saúde, Brasilia 70710-907, Brazil; alineloschinutri@gmail.com; 3Sciences and Technology in Health Program, Universidade de Brasilia, Brasilia 72220-275, Brazil; evertonsilva@unb.br (E.N.d.S.); grafbc10@gmail.com (G.F.B.C.); 4Campus Ceilândia, Universidade de Brasilia, Brasilia 72220-275, Brazil; 5Rehabilitation Sciences Program, Universidade de Brasilia, Brasilia 72220-275, Brazil

**Keywords:** cost-effectiveness analysis, COVID-19, health-related quality of life, rehabilitation, respiratory muscle training

## Abstract

Objectives: This study aims to provide model-based cost–utility estimates for the addition of inspiratory muscle training (IMT) in COVID-19 pulmonary rehabilitation (PR). Methods: A cohort model comparing IMT with PR (intervention group) to IMT with only PR (control group) was used. The payer perspective from the Unified Health System in Brazil was adopted. Effectiveness parameters: Effectiveness was measured in quality-adjusted life years (QALYs). Probabilistic sensitivity analyses were performed using 1000 Monte Carlo simulations. A beta probability distribution was assumed for utilities, and a gamma distribution was applied to the costs. A cost-effectiveness threshold of BRL 40.000/QALYs was applied. Results: As the threshold of BRL 40.000/QALYs, we obtained 512 (51.2%) simulations that can be considered cost-effective to IMT added in PR programs. IMT added in PR treatment was more expensive (USD 317.73 versus USD 293.93) and more effective (incremental utility of 0.03 to INT group) than PR alone. The incremental cost-effectiveness ratio (ICER) was 793.93 USD/QALY. Conclusions: IMT added to PR is a cost-effective alternative compared with PR for post-COVID-19 patients. This strategy may result in net cost savings and improvements in the QALYs for these patients.

## 1. Introduction

Since 11th March 2020, when the World Health Organization (WHO) declared the outbreak of COVID-19 cases to be a pandemic, several public health strategies have been implemented worldwide. One such significant problem is the prolonged persistence of symptoms in various organs after infection, which may persist for up to 12 weeks in some individuals [1]. Long COVID-19 is defined as the presence of clinical symptoms in individuals who have been infected with the virus and continue for at least three months after the onset of the disease or appear as new symptoms that persist for more than two months without any other explanation and cannot be linked to other existing health conditions [2]. To reduce the consequences of prolonged COVID-19, international guidelines strongly recommend rehabilitation programs for these individuals [3,4].

The rehabilitation program following COVID-19 comprises exercise training modalities, psychosocial counseling, and education on symptom management. A recent systematic review showed that comprehensive pulmonary rehabilitation interventions may improve dyspnea and exercise tolerance in adults with COVID-19 [5]. Among these exercise training modalities to enhance this improvement, inspiratory muscle training (IMT) has been used to enhance better results on long-term function and recovery in post-COVID-19 patients [6].

The public health system has not yet guaranteed the availability of IMT in Brazil’s Unified Health System (Sistema Único de Saúde [SUS]). The National Committee for Health Technology Incorporation in the Unified Health System (Comissão Nacional de Incorporação de Tecnologias no SUS—CONITEC), which oversees commissioning and conducting health technology assessments, including decisions on the incorporation and disinvestment of health technologies, has pointed out that economic assessments of respiratory therapy interventions are required for respiratory diseases, including COVID-19. As a result, this study aims to provide cost–utility estimates for adding IMT to the COVID-19 rehabilitation program compared with traditional COVID-19 rehabilitation programs, which may support decision-making in the SUS.

## 2. Methods

### 2.1. Study Design

A cost–utility analysis was conducted alongside a pragmatic randomized double-masked controlled trial in Brasilia, Brazil. The entire protocol has been previously described [7]. Participants were selected for the study by screening from four to six weeks after discharge from the ICU. Physicians from long-term COVID-19 ambulatory hospitals in Brazil conducted the screening process. To be eligible for the study, adult participants had a confirmed COVID-19 diagnosis that required hospitalization and were classified as moderate to severe within three months prior to the study’s recruitment [8]. Participants were excluded from the study if they met any of the following criteria: pregnancy, dependence on others for activities of daily living, diagnosed cognitive impairment or spinal cord injury, advanced neurological disorders, or terminal illness. Furthermore, individuals who were not fully prepared to engage in the prescribed treatment regimen were not eligible to participate.

The participants were randomly allocated in a 1:1 ratio to either the first intervention group (PR combined with IMT—INT Group) or the control group (PR combined with sham IMT—CON Group). Both the groups underwent the same pulmonary rehabilitation (PR) program. The PR staff prescribed individualized PR within the pre-specified parameters recommended by the PR guidelines [9]. The same physical therapists administered PR.

This study adhered to the Consolidated Health Economic Evaluation Reporting Standards 2024 [10]. The model structure was built after reviewing previously published models and considering the most conservative evidence of treatment effects related to the post-COVID-19 rehabilitation program [11].

#### 2.1.1. Ethics and Dissemination

The Brazilian National Ethics Committee (CONEP) granted approval for the project on 10 July 2020 (document number 4324069). This study was registered in the Clinical Trials Registry Platform under the identifier NCT04595097. The study participants were offered a choice to voluntarily engage in the research.

#### 2.1.2. Intervention

The study protocol involved supervised intervention with 50–60% of the maximum inspiratory pressure (MIP) as the exercise load. Participants completed two sets of 30 breaths daily, totaling 60 breaths, with a 2 min rest between sets, three times per week. A inspiratory muscle training device (POWERbreathe^®^ KH2, HaB International Ltd., London, UK) was employed. Following a previously established protocol, the inspiratory muscle training (IMT) program was combined with a pulmonary rehabilitation (PR) program over eight weeks [12].

The study subjects were asked to complete 30 quick and forceful inhalations, with short breaks allowed, if necessary, within a one-minute timeframe. The maximum inspiratory pressure (MIP) and S-Index (maximal dynamic inspiratory pressure) were adjusted weekly to keep the training intensity at the highest manageable level, ranging from 50% to 60% of the most recent MIP. Weekly MIP evaluations were conducted to ensure training intensity. All respiratory muscle strength assessments were carried out by a trained physiotherapist. For the control group, the training workload was fixed at 10% of the initial MIP and remained unchanged throughout the study. The inspiratory muscle training (IMT) protocol was implemented after pulmonary rehabilitation (PR).

Both groups performed a similar PR program. It consisted of 24 sessions over 8 weeks with three weekly sessions lasting from 30 to 50 min each. The sessions included endurance training or interval training at moderate to high intensities. The intensity was prescribed using heart rate (from the first ventilatory threshold [1stVT]) measuring up to 10% above and oxygen saturation (above 90%) as the exercise targets.

Skilled physical therapy professionals held orientation sessions to fine-tune exercise recommendations.

#### 2.1.3. Costs Analysis

The rehabilitation program’s direct medical expenses, including medical consultations, device supplies, and rehabilitation teams, were evaluated based on the public health system’s perspective for each patient. The costs were determined using real-life scenarios and were converted to American dollars (USD), with an average conversion rate of BRL 5.10 per US dollar. The applied time horizon was 12 weeks, which was the average time recommended for patients in COVID-19 rehabilitation programs [13]. The information system used for obtaining the cost data was open-access and allowed for consultation on the costs of medical procedures adopted by the Brazilian Ministry of Health [14]. However, a correction factor of 2.8 was applied to the data, as they only represent a third of the federal government’s spending.

### 2.2. Study Perspective

The adopted perspective was the one of the SUS, as the payer for public healthcare system in Brazil. This perspective was adopted to reflect decision-making related to allocating resources for adopting the technology.

### 2.3. Cost–Utility Analysis

A cost–utility analysis of the IMT compared to the CON was performed, adopting a public healthcare perspective. The outcome measure in the analysis was quality-adjusted life years (QALYs) [15]. The QALY integrates the duration and quality of life into a single metric, facilitating comparisons of efficacy between treatment groups. The findings are presented as an incremental cost-effectiveness ratio (ICER), which represents the differential in costs between the two groups divided by the difference in effects (QALYs), calculated as follows [15]:ICER = mean cost (IMT) − mean cost (CON)/mean effect (IMT) − mean effect (CON) = ΔC/ΔE

### 2.4. Measure of Effectiveness

Effectiveness was measured in quality-adjusted life years (QALYs). The utility values associated with each model state were obtained from the literature because the impact of the different conditions caused by COVID-19 on quality of life has not been published yet for the Brazilian population. It was estimated that the average disutility was −0.061 and −0.155 for patients after being discharged from the ward and ICU, respectively, resulting in average utilities of 0.724 and 0.693, respectively [16].

### 2.5. Sensitivity Analysis

A univariate sensitivity analysis was performed, including almost all model parameters except for the transition probabilities, because they are a micro-costing dependent on the individual characteristics of each established profile. Confidence intervals (CIs) were applied as lower and upper boundaries, or, alternatively, a variation of 15% was applied.

Probabilistic sensitivity analyses were performed using 1000 Monte Carlo simulations. A beta probability distribution was assumed for utilities, and a gamma distribution was applied to the costs. A cost-effectiveness threshold of BRL 40,000/QALY was applied [17].

## 3. Results

The participants in the INT and CON groups were very similar at the protocol’s baseline. Most of them were obese, diabetic, and hypertensive. The main COVID-19 symptoms were fatigue and myalgia. They were treated for respiratory failure, with oxygen therapy and invasive mechanical ventilation as the main interventions (Table 1).

Table 2 presents the composition of the treatment costs for each group. The expenses were divided by structure (IMT device treadmill, oxygen device supplier), supplies (cleaners, nose catheter, masks), and rehabilitation team salaries (physicians, physical therapists, and nurses).

Of the 1000 simulations, 122 (12.2%) resulted in a lower cost and greater effectiveness to the IMT group than the CON group; that is, they were dominant. Of the simulations that costed more and were more effective, 390 (39%) were below the BRL 40,000/QALY threshold and 103 (10.3%) were above it. In total, 80 (8%) simulations had lower cost and effectiveness than the comparator, and 305 (30.5%) had a higher cost and lower effectiveness; that is, they were dominated.

If we add the 122 dominant simulations, with the simulations that had superior cost and effectiveness but remained below the threshold of BRL 40,000/QALY, we obtained 512 (51.2%) simulations that can be considered cost-effective (Figure 1). This proportion above the average allows us to conclude that IMT addiction in post-COVID rehabilitation programs was cost-effective, given the threshold established in Brazil.

The probabilistic sensitivity analysis showed that for all simulations, IMT added in PR treatment was more expensive and more effective than PR alone (CON group) (Table 3).

## 4. Discussion

Our results show that IMT added to PR in post-COVID-19 patients is a cost-effective strategy of care. To our knowledge, this is the first study that investigated the cost–utility analysis of IMT in post-COVID-19 patients. These results may support the decision about the implementation of IMT in post-COVID-19 rehabilitation programs, aiming to bring more efficiency to the Brazilian health system.

These findings are aligned with previous studies that investigated the cost-effectiveness of PR in other chronic respiratory diseases, i.e., chronic pulmonary obstructive disease (COPD) [18,19]. However, these studies compared PR with usual care, which is a different comparator of our study. Zhang et al. demonstrated that a 3-month PR program achieved CNY 3.655 medical savings per patient per year compared to usual care for COPD patients [19]. Mosher et al. demonstrated that PR resulted in net cost savings per patient of USD 5721 and had an incremental cost-effectiveness ratio of USD 884 per session for USD 50,000/QALY [18]. Our ICER was USD 793, which was close to this ratio. Middle-income countries, like Brazil, face a significant challenge in providing equal access to healthcare due to the high treatment costs, which puts pressure on their universal health systems. The costs of the post-COVID-19 program were low, which can lead to possible incorporation into the SUS because it could be more of a benefit than the traditional PR program offered for post-COVID-19 patients.

Modest QALY gains, particularly in the general population, can be seen over one year. However, it is important to note that the QALY is an annual adjusted measure, limiting the benefits to that specific period. The chosen time horizon was for the model to prevent hospital readmission, which would mean only a few days of reduced quality of life. Although the accumulation of incremental benefits is reduced due to this minor disutility, the total impact remains significant.

Recent studies have demonstrated that IMT improved persistent symptoms and exercise tolerance in post-COVID-19 patients [20,21]. During our trial, we employed IMT in conjunction with a biofeedback device. This approach has the potential to offer additional clinical benefits, as it can influence all aspects of muscular performance, including strength, power, and work capacity [7]. Therefore, we have shown that addiction to IMT in a PR program could be implemented in a rehabilitation program offered in the Brazilian public health system.

Our investigation has limitations. As is the case with any model-based analysis, our outcomes are contingent upon the soundness of the model’s structure, the accuracy, reliability, and suitability of the data utilized for the parameters, and the comprehensiveness of the sensitivity analyses that consider all reasonable values and scenarios. One of the significant constraints was the dearth of studies that could have supplied dependable estimates on the protracted expenses of chronic COVID-19 from the Brazilian public health system’s viewpoint. It is worth mentioning that this constitutes a crucial void in the evidence base and that future investigations employing precise costing methodologies, such as microcosting, are merited. Lastly, we refrained from undertaking internal validation and instead relied on cross-validation techniques to ensure model stability.

Our study also has several advantages. As far as we know, this is the first cost–utility analysis of IM treatment on PR based on real-world data in Brazil. The model enabled us to estimate the impact of the PR program on costs and patient quality of life in a specific subpopulation, which could lead to better treatment outcomes. Although economic evaluations can help inform decision-makers and support them during the decision-making process, they should not be considered the only criterion. Other factors, such as ethics, equality, and budgetary impact, should also be considered.

## 5. Conclusions

The findings of this economic evaluation study suggest that addiction to IMT in PR post-COVID-19 rehabilitation programs among Brazilian patients may result in net cost savings and improvements in QALY. Given these findings, payers, including the government and private health insurance, should identify policies to increase access and adherence to PR programs for patients living with post-COVID-19 sequelae.

## Figures and Tables

**Figure 1 ijerph-21-01434-f001:**
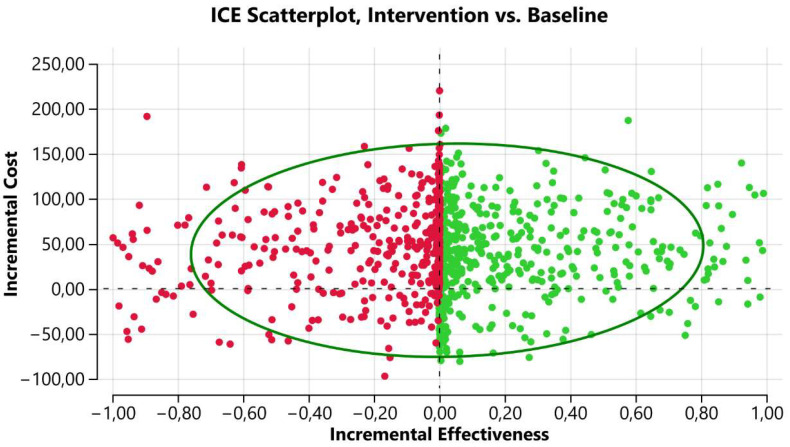
Cost–utility analysis scatterplot of INT versus CON intervention for post-COVID-19 patients.

**Table 1 ijerph-21-01434-t001:** Participant characteristics.

Descriptor	Total (n = 52)	PR-IMT Group (n = 25)	PR-Group (n = 27)
Age years (MED ± IQ 25–75)	57 (43–63)	56 (44–63)	57 (42–63)
BMI-kg/m^2^ (MED ± IQ 25–75)	30.9 (27.3–33.4)	30.5 (27.5–33.5)	31.1 (27–33.3)
Hospitalization days (MED ± IQ 25–75)	23.5 (12–40.5)	25 (13–32)	22.5 (11.5–45.5)
Risk Factors			
Arterial hypertension (n-%)	36 (69.2%)	16 (64%)	20 (74.1%)
Diabetes (n-%)	15 (28.8%)	8 (32%)	7 (26.9%)
Cardiovascular disease (n-%)	9 (17.3%)	4 (16%)	5 (18.5%)
Pulmonary disease (n-%)	2 (3.8%)	1 (4%)	1 (3.7%)
Medications in Use			
None (n-%)	8 (15.3%)	5 (20%)	3 (11.1%)
Antihypertensive (n-%)	34 (60.7%)	15 (60%)	19 (70.3%)
Glucose control (n-%)	18 (34.6%)	10 (40%)	8 (29.62%)
Anticoagulant (n-%)	6 (11.5%)	2 (8%)	4 (14.81%)
Post-COVID Symptoms			
Headache (n-%)	21 (40.3%)	9 (36%)	12 (44.4%)
Myalgia (n-%)	29 (55.7%)	15 (60%)	14 (51.8%)
Fatigue (n-%)	33 (63.4%)	17 (68%)	16 (59.2%)
Chronic cough (n-%)	10 (19.2%)	5 (20%)	5 (18.5%)
ARF Treatment			
Oxygen therapy (n-%)	18 (34.6)	9 (36)	9 (33.3)
Noninvasive ventilation/HFOT (n-%)	13 (25)	6 (24)	7 (25.9)
Invasive mechanical ventilation (n-%)	21 (40.4)	10 (40)	11 (40.7)

Legends: PR—pulmonary rehabilitation; IMT—inspiratory muscle training; BMI—body mass index; ARF—acute respiratory failure; HFOT—high-flow oxygen therapy.

**Table 2 ijerph-21-01434-t002:** Costs applied in the cost–utility analysis.

Type of Cost	Type of Resources	Units	Monetary Values (USD)	Sources of Information
Structure	IMT device	1 unit	1200	Market price
Structure	Treadmill	1 unit	1150	Market price
Structure	Resistance exercise station	2 units	1132	Market price
Structure	Oxygen supplier device	2 units	960	Market price
Rehabilitation team salaries	Physicians, physical therapists, nurses, and administrative employment	Per month	2347	Health Secretary of Distrito Federal
Supplies	Cleaners, nose catheter, individual protective equipment	200 units	1345	SIGTAP

Legends: IMT: inspiratory muscle training; USD: American dollars; SIGTAP: Sistema de Gerenciamento da Tabela de Procedimentos, Medicamentos do Sistema Único de Saúde (SUS)—Brazil.

**Table 3 ijerph-21-01434-t003:** Incremental cost–utility analysis with INT/control groups.

Groups	Treatment Cost (US$)	QALY	Incremental Cost (US$)	Incremental Utility	ICER (USD/QALY)
INT	317.73	0.23	23.80	0.03	793.33
CON	293.93	0.20			

Legend: IMT: inspiratory muscle training; CON: control group; QALY: quality-adjusted life years; ICER: incremental cost-effectiveness ratio; USD: American dollars.

## Data Availability

All data is available upon request.

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
