# Peer review of "Cost–Utility Analysis of Supervised Inspiratory Muscle Training Added to Post-COVID Rehabilitation Program in the Public Health System of Brazil"

_ijerph, 2024, doi:10.3390/ijerph21111434_

Round 1
Reviewer 1 Report
Comments and Suggestions for Authors
Cost-utility Analysis of Supervised Inspiratory Muscle Training added to post Covid Rehabilitation Program in the Public health System in Brazil.
First of all, thank you very much for your contribution. I think it is a very interesting study that can contribute to science but there are some things I would like to recommend to improve the quality of the manuscript.
1.- In the introduction I suggest that you state what kind of persistent symptoms and their frequency in order to link them later to the pulmonary rehabilitation treatment.
2.- You refer to the improvements of IMT treatment in the function of patients, please explain a little more what this type of technique consists of and more specifically where these improvements are reflected in the long term in patients affected by post-COVID.
3.- In the design section of the study you refer to people dependent on others in their activities of daily living as exclusion criteria, how did you measure that, and what was the reason for excluding these dependent people if this condition mostly affects the independence of the affected people and would also be a valuable measure to reflect on the improvement of those affected by post-COVID syndrome. Most of the people admitted to the ICU have after-effects on their daily life.
4.- Regarding the measure of efficacy, you refer that no quality-of-life results have been published in the Brazilian population, it would be advisable to reflect and take as a reference in other regions or populations.
5.- Characteristics of the participants that are not detailed above are explained in the results. It is not entirely clear to me how the cost-effectiveness of the therapy was measured.
6.- The debate refers to the improvement of IMT on persistent symptoms, such as strength, power or work capacity, so it could be implemented in the health system, was this measurement not done? I understand that it is referred to with some subjectivity and cannot be attributed with proven evidence. This would be a major limitation of the study.
Author Response
2. Point-by-point response to Comments and Suggestions for Authors |
Comments 1: [In the Abstract, Please avoid using abbreviations alone unless you provide what they stand for especially when they first appear in text. Per say, please edit both (INT group), (CON group), and QALY. Keywords: it is recommended to arrange keywords in an alphabetical order.
|
Response 1: We updated the abstract according to reviewer comments. All changes were highlighted in yellow.
. |
Comments 2: Please define recruitment and sampling method. Also, add information related to ethical approval (name of institution obtained the approval from, number and date of approval). Were the participants consented to participate in this study? please add consent statement. Please provide a description of the intervention provided in the intervention group; the inspiratory muscle training (IMT), such as intensity of intervention (frequency, duration, procedure or protocol, exercises, activities performed, etc.)
|
Response 2. We included in the Methods Section information about Ethics Committee and intervention (highlighted in yellow). The recruitment and sampling methos were in the Methods Section (Study Desgin sub section, highlighted/yellow). We included the consent form applied in the participants as Supplemntary Materials.
Comments 3: Results section needs expansion. I could not find both table 1 and table 2 in the manuscript (missing). Also, participants demographics (i.e., age, gender, sings and symptoms, diagnosis, etc.) at baseline are not reported (please add them in a table).
Response 3: We included the tables 1 and 2 in the revised Manuscript, as weel as the Figure 1 in the Results Section.
|
Reviewer 2 Report
Comments and Suggestions for Authors
Dear Authors,
Thank you for your submission.
This is a well-written, informative, and well-implemented study.
I have only few review comments, suggestions, and recommendations for your consideration as follows:
Abstract
Please avoid using abbreviations alone unless you provide what they stand for especially when they first appear in text. Per say, please edit both (INT group), (CON group), and QALY.
Keywords: it is recommended to arrange keywords in an alphabetical order.
Methods
Please define recruitment and sampling method. Also, add information related to ethical approval (name of institution obtained the approval from, number and date of approval). Were the participants consented to participate in this study? please add consent statement.
Please provide a description of the intervention provided in the intervention group; the inspiratory muscle training (IMT), such as intensity of intervention (frequency, duration, procedure or protocol, exercises, activities performed, etc.)
Results
Results section needs expansion. I could not find both table 1 and table 2 in the manuscript (missing). Also, participants demographics (i.e., age, gender, sings and symptoms, diagnosis, etc.) at baseline are not reported (please add them in a table).
Please make the required edits and I will be happy to look at the revised manuscript.
Wishing you all the best in your current and future scholar work.
Many thanks.
Author Response
Comments 1: [In the Abstract, Please avoid using abbreviations alone unless you provide what they stand for especially when they first appear in text. Per say, please edit both (INT group), (CON group), and QALY. Keywords: it is recommended to arrange keywords in an alphabetical order.
|
Response 1: We updated the abstract according to reviewer comments. All changes were highlighted in yellow.
. |
Comments 2: Please define recruitment and sampling method. Also, add information related to ethical approval (name of institution obtained the approval from, number and date of approval). Were the participants consented to participate in this study? please add consent statement. Please provide a description of the intervention provided in the intervention group; the inspiratory muscle training (IMT), such as intensity of intervention (frequency, duration, procedure or protocol, exercises, activities performed, etc.)
|
Response 2. We included in the Methods Section information about Ethics Committee and intervention (highlighted in yellow). The recruitment and sampling methos were in the Methods Section (Study Desgin sub section, highlighted/yellow). We included the consent form applied in the participants as Supplemntary Materials.
Comments 3: Results section needs expansion. I could not find both table 1 and table 2 in the manuscript (missing). Also, participants demographics (i.e., age, gender, sings and symptoms, diagnosis, etc.) at baseline are not reported (please add them in a table).
Response 3: We included the tables 1 and 2 in the revised Manuscript, as weel as the Figure 1 in the Results Section.
|
Round 2
Reviewer 1 Report
Comments and Suggestions for Authors
The manuscript is now suitable for publication
Author Response
our manuscript has now been checked by our external academic editor. There
is one concern that needs to be addressed and corrected. There is 39% overlap
with other published studies. This is mostly due to standard sentences in
methods etc, but please rephrase sentences so that we can reduce similarity
rate and proceed with processing you manuscript.
Answer: We changed the manuscript to avoid the overlap. We submitted a new file with changes mainly in Methods Section .
